# Anterior Screw Insertion Results in Greater Tibial Tunnel Enlargement Rates after Single-Bundle Anterior Cruciate Ligament Reconstruction than Posterior Insertion: A Retrospective Study

**DOI:** 10.3390/medicina59020390

**Published:** 2023-02-17

**Authors:** Yangang Kong, Lifeng Yin, Hua Zhang, Wenlong Yan, Jiaxing Chen, Aiguo Zhou, Jian Zhang

**Affiliations:** 1Department of Orthopaedics, The First Affiliated Hospital of Chongqing Medical University, No. 1 Youyi Road, Yuzhong District, Chongqing 400016, China; 2Orthopedic Laboratory, Chongqing Medical University, Chongqing 400016, China

**Keywords:** ACL reconstruction, interference screw, tibial tunnel enlargement, screw position, MRI

## Abstract

*Background and Objectives*: Tunnel enlargement (TE) is a widely reported phenomenon after anterior cruciate ligament reconstruction (ACLR). Given the paucity of knowledge in the literature, it remains unclear whether screw position in the tunnel affects TE. This retrospective cohort study evaluated differences in postoperative tunnel enlargement rates (TER) and clinical results between anterior and posterior tibial interference screw insertion during single-bundle ACLR using autologous hamstring grafts. *Materials and Methods*: A group of consecutive patients that underwent primary arthroscopic single-bundle ACLR in our hospital were screened and divided into two groups based on the position of the tibial interference screw (determined by Computer Tomography within 3 days after surgery): anterior screw position group (A) and posterior screw position group (B). The bone tunnel size was measured using magnetic resonance imaging (MRI) performed 1 year after surgery. International Knee Documentation Committee (IKDC) score and the Knee Injury and Osteoarthritis Outcome Score (KOOS) were used for clinical results 1 year postoperatively. *Results*: 87 patients were included. The TER of Group A is higher than that of Group B (43.17% vs. 33.80%, *p* = 0.024). Group A showed a significant increase (12.1%) in enlargement rates at the joint line level than group B (43.77% vs. 31.67%, *p* = 0.004). Moreover, KOOS and IKDC scores improved in both groups. There were no significant differences in clinical outcomes between the two groups. *Conclusions*: One year after ACLR, patients with posterior screw showed significantly lower TE than patients with anterior screw. However, the position of screw did not lead to differences in clinical results over our follow-up period. Posterior screw position in the tibial tunnel maybe a better choice in terms of reducing TE. Whether the different screw positions affect the long-term TE and long-term clinical outcomes needs to be confirmed by further studies.

## 1. Introduction

Anterior Cruciate Ligament (ACL) injury is common among people who take part in competitive sports [1]. Anterior cruciate ligament reconstruction (ACLR) is an effective method to restore knee joint function and reduce the risk of joint degeneration [2,3]. Especially for adolescents and people engaged in physical work, surgical treatment is the cost-effective option [4,5]. Many factors can influence the prognosis of ACLR, such as type of graft, surgical technique, rehabilitation protocol, etc. For example, when adolescents choosing allografts, the incidence of graft re-fracture was influenced by donor’s age and sex [6]. Over the past three decades, significant advances in ACLR surgery have led to the majority of patients reporting good function following ACLR. Despite these advances, tunnel enlargement (TE) is still a widely reported phenomenon [7,8] after ACLR. It is well-established that TE mainly occurred in the first 6 months postoperatively and lasts about 2 years [9,10].

Although TE may not affect short-term clinical outcomes, graft healing in the tunnel may be affected after reconstruction surgery. TE after 6 months postoperatively may indicate poor graft-to-bone healing, resulting in increased laxity which may ultimately affect the function of knee joint [11,12]. TE is a common phenomenon, but little is known about relevant mechanisms. Synovial fluid, graft, and certain cytokines are main biological factors [13,14,15]. Mechanical factors, such as micromotion between tunnel and graft, fixation, bone tunnel location, may also be critical [16]. Accordingly, TE is a complex process.

Interference screws are commonly used for graft fixation on the tibial side by eccentric compression of graft tendons. The interference screw is usually placed under or beyond the graft. However, to the best of our knowledge, there is a paucity in literature regarding the influence of interference-screw position on postoperative TE and postoperative clinical results. Considering the significant role of the ACL in the anterior and posterior stabilization of the knee, these two screw different positions might influence the biomechanical relationship between graft and bone tunnel, leading to differences in postoperative TE and clinical outcomes.

Therefore, this study aimed to verify whether the position of the tibial compression screw impacts postoperative tunnel enlargement rates (TER) and clinical outcomes. The hypothesis was that different insertion locations could affect the postoperative tibial tunnel size and clinical outcomes following ACLR.

## 2. Materials and Methods

### 2.1. Participants

The research was approved by the Ethics Committee of The First Affiliated Hospital of Chongqing Medical University (number: 2021-360). We retrospectively collected data of consecutive patients undergoing primary ACL reconstruction with hamstring graft between January 2016 and January 2021 in our institution. Dr. Hua Zhang screened our patients into the retrospective cohort based on the inclusion/exclusion criteria of the study. The inclusion criteria: (1) patients older than 18 years old; (2) patients who underwent primary single bundle ACLR using autologous hamstring graft; (3) a documented tibial tunnel diameter on the patient’s medical record; (4) patients with a follow-up of at least 1 year; (5) patients with Computer Tomography (CT) imaging within 3 days of surgery; (6) patients with Magnetic resonance imaging (MRI) data of ipsilateral knee one year after surgery. Patients were excluded for the following reasons: (1) patients with other ligament injuries simultaneously, like medial collateral ligament (MCL); (2) patients previously suffered from surgery or trauma regarding the affected knees; (3) patients’ lack of Computer Tomography (CT) imaging within 3 days after surgery; (4) patients’ lack of MRI or clinical data 1 year after surgery. Patients (*n* = 56) were excluded for the following reasons: lack of CT within 3 days after surgery (*n* = 3), lack of MRI 1 year after surgery (*n* = 24), lack of clinical data (KOOS, IKDC, etc.) 1 year after surgery (*n* = 13), tunnel diameter was not recorded during the operation (*n* = 2), and complicated with other ligament injuries and reconstruction (*n* = 14). Ultimately, 87 patients were included in the study (Figure 1).

All the enrolled patients were divided into two subgroups based on the CT-scan which are performed routinely within 3 days after surgery. The tibial tunnel was divided into two parts by a line perpendicular to the anteroposterior axis of the tibia. A line parallel to the posterior condylar line was made through the center of the tibial tunnel. If most of the screws (>50%) were in front of the line, they were attributed to group A (anterior); otherwise, they were attributed to group B (posterior) (Figure 2). Included patients were divided into two groups: the anterior screw position group (A) (*n* = 51) and the posterior screw position group (B) (*n* = 36).

### 2.2. Surgical Technique

All patients included in this study underwent ACL reconstruction with suspension fixation devices at the femoral side and tibial aperture fixation by absorbable interference screw (DePuy Mitek MILAGRO BR) and reinforced by staples (Arthrex Spiked Ligament Staple) by the same surgeon. First, diagnostic arthroscopy was used to confirm ACL rupture. We prepared tendons as previously described by Chiang et al. [17]. The two tendons were treated with three or two folds to form four- or five-strand grafts.

Meanwhile, the femoral tunnel was opened at the anatomic site of the ACL through an anterior medial (AM) portal. The bone tunnel size was the same as graft tendon’s test diameter. The length of the implant of the femoral tunnel was at least 20 mm in all patients. The graft was fixed by Endobuttons (Smith&Nephew) and 8 mm was over-drilled for button flip.

The tibial tunnel was drilled at the ACL tibial footprint. The tunnel diameter was created to be the same as the graft tendon’s test diameter. After the graft was fixed at the femoral side, circulatory loading was performed to ensure that the graft was not impacted. Then, the tibial tunnel graft was fixed in a straight knee position using interference screws (DePuy Mitek MILAGRO BR) (Figure 3) of length 25 mm, which matched the tunnel diameter, then reinforced by a staple (Arthrex Spiked Ligament Staple) outside. Intercondylar fossa plasty was performed according to the doctors’ intraoperative decision.

### 2.3. Rehabilitation

After the patients awakened from anesthesia, active quadriceps isometric exercise and range-of-motion (ROM) exercise were initiated with an ACL limited-motion brace. The knee flexion was limited to 0°to 90° for the first month. Knee flexion increased to 120 degrees in the second month. Then, a full range of motion of the knee is possible. Proprioception exercises was included in the rehabilitation protocol for all our patients. Patients with a simple cruciate ligament injury can be fully weight-bearing. We recommend that patients with meniscal injuries undertake weight-bearing training 6 weeks later. The others perform proprioception exercises (four directional balance exercises on a single leg, 30 s heel walking, 30 s toe walking, walking a straight line with eyes closed, walking backwards with eyes open) at least once daily. Patients with meniscal repaired were required to use a pair of crutches (non-weight-bearing in the first 4 weeks, partial weight-bearing (<15% weight) between 4 to 6 weeks, fully weight-bearing after 6 weeks). All patients returned to normal walking 2 months after surgery. Return to competitive sports was dependent on patient’s condition 6 months to 1 year after the reconstruction.

### 2.4. Imaging Evaluations

Within 3 days after ACLR surgery, each patient routinely underwent a CT scan to determine the relative position of the graft and the screw (Figure 2). At 1 year follow-up, the patient underwent MRI scan. The reason why we chose MRI instead of CT was that CT has radiation exposure and MRI can be used to assess graft healing. An increasing body of evidence suggests that MRI yields good performance for measuring the dimensions of bone tunnels [7,18,19,20,21]. Although MRI is less precise than CT, MRI provides more information about the status of the reconstructed ligament and the condition of the knee than CT and X-ray [22].

MRI (1.5T) was performed 12 months after surgery. Dr. Jiaxing Chen divided the patients into two subgroups based on the CT-scans that are performed routinely within 3 days after surgery. Two experienced orthopedists (Dr. Yangang Kong and Dr Lifeng Yin) measured the tunnel size on MRI. They were independent of each other, and all the processes were supervised by Prof Jian Zhang and Prof Aiguo Zhou. Tunnel size would be measured again by Dr. Yangang Kong and Dr Lifeng Yin on MRI after 6 weeks. The intraclass correlation coefficient (ICC) was evaluated based on the above data that they measured. The final TER was obtained by the average of both values.

Measurements of the tunnel were based on methods described in the previous literature [9]. The anteroposterior tunnel size was determined at the joint line, center level of the tunnel without screw engagement and top of the screw on sagittal images. The anterior–posterior distance perpendicular to the long axis was recorded as the tunnel width of each level (Figure 4).

TER was calculated as follows: TER = (MRI measured diameters − Drilled diameters)/(Drilled diameters) × 100% [23].

All measurements were performed on the picture archiving and communications system (PACS) using automated distance calculation tools.

### 2.5. Clinical Evaluations

The preoperative Lachman test was performed by the surgeon and the results recorded in the medical record. The postoperative Lachman test was performed by the same surgeon in the outpatient department 1 year after ACLR, and the results were recorded in the follow-up data. Functional scores, including the subjective international knee documentation committee score (IKDC) and knee injury and osteoarthritis outcome score (KOOS), were also recorded. Patients’ IKDC and KOOS scores were recorded on the ward before ACLR, and postoperative functional scores were obtained in the outpatient department 1 year after surgery.

### 2.6. Statistical Analysis

Means and standard deviations were used to describe continuous quantitative data, while frequencies and proportions were used to describe qualitative data. Analysis was performed using SPSS software (SPSS 25.0; IBM, Armonk, NY, USA), and statistical significance was assumed at *p* < 0.05. Two independent sample *t*-tests and χ^2^ tests were performed to evaluate differences between the group A and the group B. The interobserver consistency of main TER data between the two observers was evaluated by calculating ICC values.

### 2.7. Sample Size

Post-hoc analysis was performed using the power analysis sample size PASS software (PASS 15.0.5; NCSS, LLC, Kaysville, UT, USA). For a total sample size of 87 and type I error (α) of 0.05, the study was expected to provide a power (1-β) of 0.88.

## 3. Results

A total of 87 patients were included in this study. The mean follow-up was 14.5 months (Range 12–19 months, group A 14.86 ± 1.96, group B 15.33 ± 2.02, *p* = 0.282). There were no significant differences between the two groups in age, gender, BMI, graft length, intraoperative tibial tunnel size (diameter and length), and meniscus tear (Table 1).

### 3.1. MRI Results

Eventually, we calculated the intraclass intraobserver and interobserver correlation coefficients, which showed that our measurement method is reliable (Table 2). The average tibial TERs of the group A and the group B was 43.17% and 33.80%, respectively. Compared with the patients in the group B, the patients in the group A had greater TER at the level of the joint line, center of the tunnel without screw engagement and top of interference screw. However, group A showed significantly increased TER only at the joint line level than group B (43.77% and 31.67%, *p* = 0.004). Overall, at one year postoperatively, group A showed significantly greater TER than group B (*p* = 0.024) (Table 3).

### 3.2. Clinical Results

Groups A and B showed significant improvement in KOOS and IKDC scores. No significant difference in the IKDC and KOOS was found between the two groups (Table 4). Importantly, both groups had significantly decreased postoperative knee laxity measured by the Lachman test (*p* < 0.01). No significant difference in the Lachman test was found between the two groups (*p* = 0.687) (Table 4).

## 4. Discussion

The most important finding of this study is that compared to posterior insertion, anterior interference screw insertion in the tibial tunnel is associated with higher tibial TER one year after primary single-bundle ACLR using hamstring autograft, especially at the level of the joint line, while there was no significant difference in clinical outcomes between the two groups.

TE of the femur and tibia have been extensively documented after ACLR [24,25,26]. Tunnel widening usually occurs in the first 6 months after surgery and becomes stable within a year, and the most significant widening occurs within the first 6 weeks and reportedly progresses until two years [10,27]. Some evidence [9,23,28,29,30,31] substantiates that the incidence of TE was 29–100% on the tibial side and 25–100% on the femoral side. It is widely accepted that the etiology of TE is multifactorial, including mechanical forces and biological factors [11,32,33,34,35]. The host immune response to the graft is a major biological factor [36]. Tunnel position, fixation method, graft motion, and the initial tension of graft are thought to be related to mechanical factors [11,37,38].

Interference screws can be exploited to obtain more stable initial fixation [38]. Wang JH et al. noted that interference screws containing β-tricalcium phosphate, which reduced TE near the screws compared with pure poly-L-lactic acid (PLLA) screw [39]. Agarwal S et al. found that a higher slope tapered interference screw could cause less damage to graft fibers, reducing graft laxity and increasing fixation stability compared to a low slope tapered interference screw [40]. However, it remains unclear whether the relative position between interference screw and graft can affect TE.

Our study showed that anterior screw insertion could result in larger tibial tunnel widening than posterior screw insertion. The graft inside the tibial tunnel was eccentric fixation in our study. The fact that we put interference screw at the front position may cause posterior shifting of the graft, which may decrease ligament tension end with more frequent micromovement during knee flexion and extension.

It has been reported that during knee joint flexion and extension, the force exerted on the graft mainly produces two effects, namely the “bungee effect” and “windshield-wiper effect” [41]. The former is described as the longitudinal movement of the implant in the bone tunnel after ACLR, while the latter is a transverse movement relative to the bone tunnel [42]. M. Jagodzinski et al. [43] demonstrated that pressure on the tunnel wall caused by graft-tunnel micromotion of the transverse axis and increased as it approached the articular surface, explaining why there are differences in TE between the two groups in our research. In the study of Jagodzinski, more significant tibial tunnel widening was observed in the sagittal plane than in the coronal plane. The “redirecting forces” at the articular entrance of the tibial tunnel depends on the graft tunnel angle and intra-ligament tension [43], providing evidence that biomechanical forces play a key role in postoperative tunnel expansion. It should be noted that different positions of interference screws may change the biomechanical environment around the graft. However, no research has hitherto assessed the influence of screw position on the biomechanical environment. We have reason to believe that posterior screw insertion can keep graft tension and reduce graft micromovement, which leads to less TE.

On the other hand, we only found significant differences in TE between the two groups at the level close to the articular surface of tibia. There is evidence suggesting that synovial fluid plays an essential role in TE, especially near the articular surface of tibia. During the healing process of graft and bone, inflammatory mediators are involved. These inflammatory mediators mainly enter the bone tunnel through synovial fluid [44]. After ACLR, higher concentrations of inflammatory factors are detected in the synovial fluid, including Interleukin-1β (IL-1β), IL-6, Tumor Necrosis Factor-α (TNF-α), nitric oxide (NO) [14,45], which can cause osteolysis [46]. Some of the literature substantiate that a high level of the above substances in synovial fluid may be involved in the occurrence of TE [47,48]. The graft-to-bone healing process may be affected by a phenomenon called the “synovial bath effect” [49]. In our study, the TER was higher in the anterior group than in the posterior group at the joint line. We hypothesize that this difference may be due to less graft tension. Hence, synovial fluid is more easily moved inside the tibial tunnel, enhancing TE near the articular surface of tibia. This needs further study to be proven.

Increased joint fluid after surgery may infiltrate between the graft and bone tissue, resulting in poor healing of the graft to the corresponding tunnel area, creating a gap called “dead spaces” [50]. The fixation point away from the articular side exit of the tunnel would make the graft unable to fill the bone tunnel as much as possible and more micromotions would be generated between the graft and tunnel, thus prolonging the healing period of the reconstructed ligament with the bone tunnel leading to TE [51,52]. This study only used 25 mm interference screws and it should be further warranted whether the insertion screw length can affect TE.

However, the clinical outcomes were comparable between the two groups. Both groups showed significant improvement in IKDC scores and KOOS scores, which is in accordance with previous literature [24,53,54,55]. Our data also demonstrated no correlation between TER and clinical outcome scores, which may be attributed to the relatively short follow-up period. Hence, long-term follow-up studies are needed in future studies.

In our research, during a comparison of two different screw insertion approaches, we noticed that posterior interference screw insertion leads to less TE at the tibial tunnel near the articular surface of tibia. However, no significant differences in clinical outcomes were found between both groups. Interestingly, it has been reported that TE at time points greater than 6 months indicates poor graft-to-bone healing, which may lead to increased laxity over time [11]. For better graft healing and revision condition, posterior screw insertion was recommended. Future mechanistic studies of how the screw placement position leads to the different microenvironments of the graft are warranted.

## 5. Conclusions

One year after ACLR, patients with posterior screw showed significantly lower TE than patients with anterior screw. However, the position of screw did not lead to differences in clinical results over our follow-up period. Posterior screw position in the tibial tunnel maybe a better choice in terms of reducing TE. Whether the different screw positions affect the long-term TE and long-term clinical outcomes needs to be confirmed by further studies.

## 6. Limitations

This is a retrospective study. Patients consisted mainly of men, although mounting evidence suggests that, for various reasons, ACL injuries occur more frequently during sports in women compared to men [1,56,57]. Moreover, CT imaging of our patients at 1 year was not available. Indeed, it is well established that computed tomography is a more accurate tool to evaluate TER, especially 3-dimensional CT because it measures the volumetric dimensions of the tunnels. Unlike CT, MRI cannot reconstruct the arbitrary interface with the corresponding software, and the image slice we selected and the measured tunnel diameter may be less accurate. In addition, no stratification was conducted based on the positions (left vs. right) of screws in the tunnel when collecting data. It remains unclear whether screw placement on the left/right side of the tunnel affects TE. Furthermore, we did not consider the distance between the tip of the screw and the articular surface. Indeed, given our study’s relatively short follow-up time, the long-term changes in the tibial tunnel could not be observed. Finally, the power (1 − β) of this study was lower than 0.9 (0.82), which may be attributed to our insufficient sample size.

## Figures and Tables

**Figure 1 medicina-59-00390-f001:**
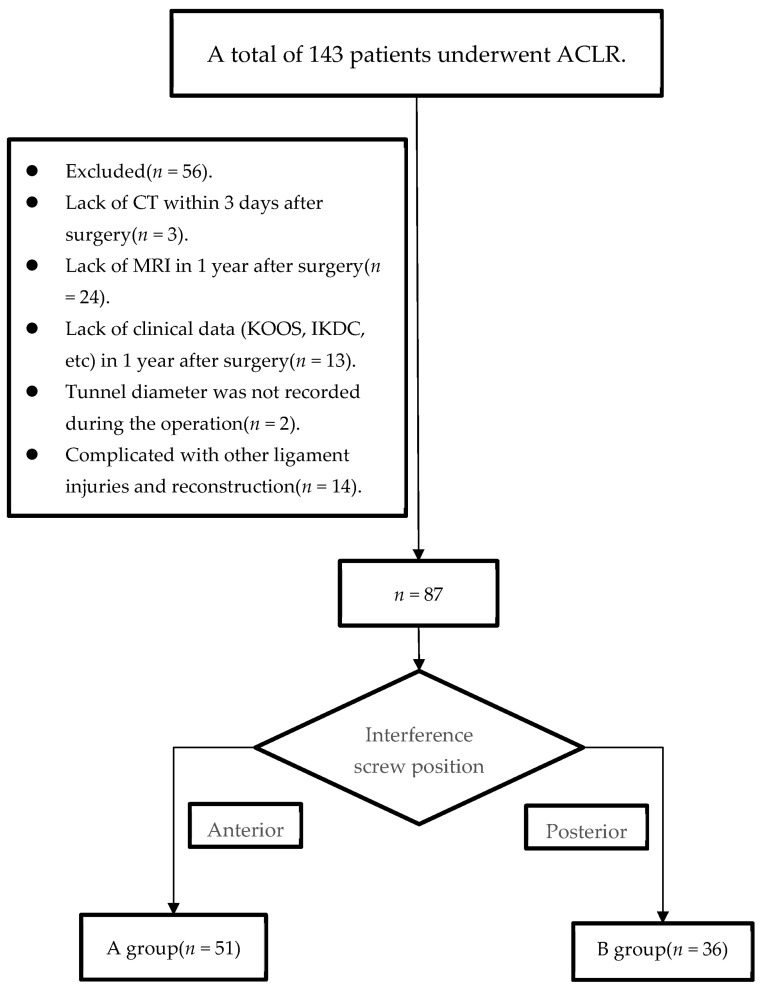
Patient flowchart. (ACLR, anterior cruciate ligament reconstruction; KOOS, knee injury and osteoarthritis outcome score; IKDC, international knee documentation committee score; CT, computed tomography).

**Figure 2 medicina-59-00390-f002:**
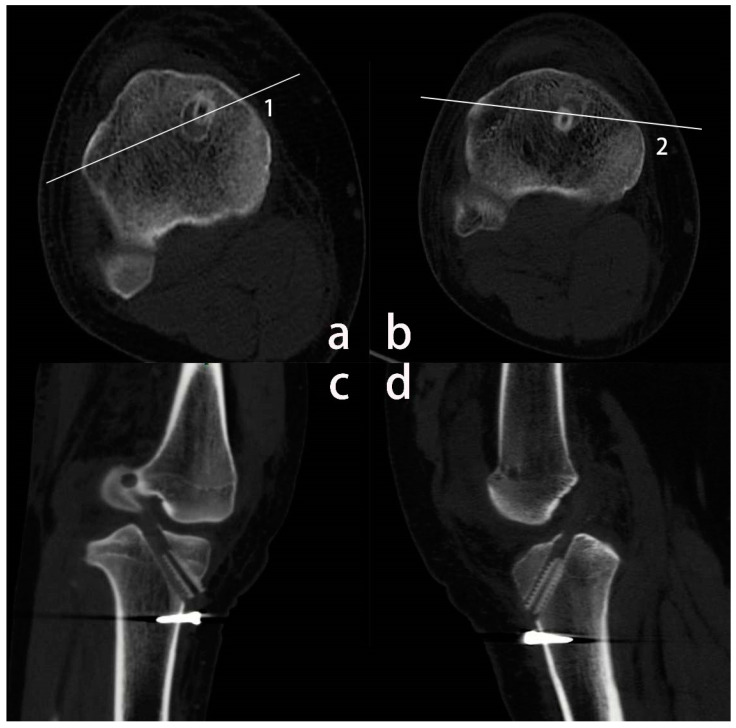
Knee CT imaging within 3 days after the operation. (**a**–**d**) belong to different patients, respectively. Line 1 and 2 are made through the center of the tibial tunnel and parallel to the posterior condylar line. (**a**), most of the screws (>50%) are in front of the line 1. (**a**,**c**) show that the patient’s tibial interference screw is located in the front of the tunnel. (**b**), most of the screws (>50%) are behind the line 2. (**b**,**d**) show that the screw is at the back of the bone tunnel.

**Figure 3 medicina-59-00390-f003:**
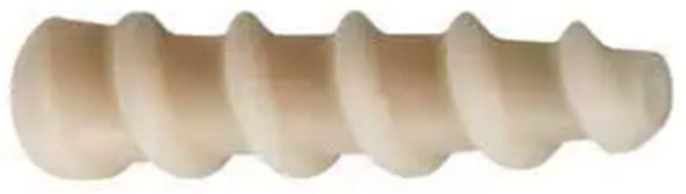
Tibial tunnel interference screw (DePuy Mitek MILAGRO BR).

**Figure 4 medicina-59-00390-f004:**
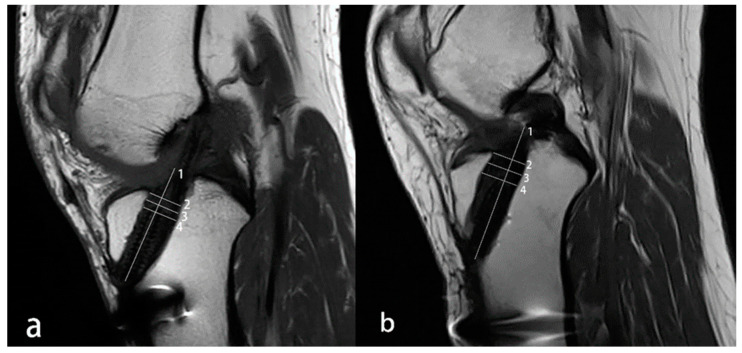
Sagittal T1-weighted MRI was used to measure the anteroposterior diameter of the tibial tunnel. (**a**), the patient’s tibial interference screw is located in the front of the tunnel. (**b**), the screw is at the back of the bone tunnel. Line 1 is parallel to the bone tunnel, and lines 2, 3, and 4 are perpendicular to line 1. Line 2 is located at the level of the joint line, line 4 is at the level of the top of the screw, and line 3 is in the middle of lines 2 and 4. The tunnel enlargement rate was calculated as follows: TER (joint line level, center level, top of screw level) = (line 2, 3 or 4 − Drilled diameters)/(Drilled diameters) × 100%.

**Table 1 medicina-59-00390-t001:** Patient Demographic Characteristics.

	Group A	Group B	*p* Value
Patients, *n*	51	36	
Age, yr	29.3 ± 9.1	29.5 ± 9.1	0.929
Sex, *n*	44 male and 7 Female	29 male and 7 female	0.475
BMI (kg/m^2^)	23.84 ± 2.23	23.82 ± 3.28	0.970
MRI follow-up, mo	14.86 ± 1.96	15.33 ± 2.02	0.282
Tibial tunnel diameter, mm	8.1 ± 0.6	8.1 ± 0.7	0.775
Tibial tunnel length, mm	43.8 ± 4.8	42.4 ± 4.5	0.175
Graft length, mm	83.1 ± 10.4	81.9 ± 10.5	0.593
Meniscus tear, *n*	26	17	0.730
Meniscus suture, *n*	19	13	0.913

Data are presented as mean ± standard deviation; Group A, anterior screw position group, Group B, posterior screw position group.

**Table 2 medicina-59-00390-t002:** ICC of inter-observer and intra-observer errors in tibial tunnel diameter measurements of different levels.

Cutting Levels	Interobserver	Intraobserver	
		1	2
joint line level	0.897	0.922	0.891
center level	0.881	0.870	0.883
screw top level	0.898	0.919	0.926

**Table 3 medicina-59-00390-t003:** Tunnel Enlargement Rate of three cutting levels and their average value on sagittal MRI imaging one year post operation.

	Group A	Group B	*p* Value
TER-joint line (SD), %	43.77 ± 18.90	31.67 ± 19.05	0.004
TER-center (SD), %	43.99 ± 18.70	34.64 ± 21.08	0.032
TER-screw top (SD), %	41.75 ± 20.72	35.11 ± 22.33	0.157
TER-average (SD), %	43.17 ± 18.33	33.80 ± 19.30	0.024

TER, Tunnel Enlargement Rate.

**Table 4 medicina-59-00390-t004:** Clinical outcome.

	Group A			Group A			*p* Value for A vs. B, Postop
	Preop	Postop	*p* Value	Preop	Postop	*p* Value	
Lachman test (+)	43/51	2/51	*p* < 0.01	28/36	3/36	*p* < 0.01	0.687
IKDC	62.22 ± 12.78	83.75 ± 7.75	0.012	61.58 ± 15.19	80.78 ± 6.08	0.029	0.058
KOOS							
Symptoms	73.79 ± 15.59	88.75 ± 6.51	*p* < 0.01	75.30 ± 13.48	87.19 ± 6.31	*p* < 0.01	0.268
Pain	84.03 ± 11.39	97.89 ± 1.47	*p* < 0.01	83.77 ± 11.60	97.62 ± 1.86	*p* < 0.01	0.460
ADL	84.95 ± 10.00	96.67 ± 1.93	*p* < 0.01	84.38 ± 9.86	96.93 ± 1.64	*p* < 0.01	0.519
Sports	46.23 ± 20.78	77.82 ± 11.39	*p* < 0.01	41.78 ± 17.55	80.48 ± 10.27	*p* < 0.01	0.268
QOL	28.44 ± 11.72	76.99 ± 12.96	*p* < 0.01	26.67 ± 11.86	77.39 ± 15.54	*p* < 0.01	0.897

IKDC, International Knee Documentation Committee (2000 IKDC SUBJECTIVE KNEE EVALUATION FORM); KOOS, Knee Injury and Osteoarthritis Outcome Score; ADL, activities of daily living (function); QOL, quality of life; Postop, postoperative; Preop, preoperative.

## Data Availability

The data associated with the paper are not publicly available but are available from the corresponding author upon reasonable request.

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
