# Peer review of "Anterior Screw Insertion Results in Greater Tibial Tunnel Enlargement Rates after Single-Bundle Anterior Cruciate Ligament Reconstruction than Posterior Insertion: A Retrospective Study"

_medicina, 2023, doi:10.3390/medicina59020390_

Round 1

Reviewer 1 Report

This manuscript aims to evaluated differences in postoperative tunnel enlargement rates (TER) and clinical outcomes at 1-year follow-up between anterior and posterior tibial interference screw insertion during single-bundle ACLR using autologous hamstring grafts. A group of consecutive patients that underwent primary arthroscopic single-bundle ACLR in our hospital were screened and divided into 2 groups based on the position of the tibial interference screw (determined by Computer Tomography within 3 days after surgery): anterior screw position group (A) and posterior screw position group (B). The bone tunnel size was measured using magnetic resonance imaging (MRI) performed 1 year after surgery. International Knee Documentation Committee (IKDC) score and the Knee Injury and Osteoarthritis Outcome Score (KOOS) were used for clinical results 1 years postoperatively.

I read the article with interest, the title is well thought out and faithfully reflects the content of the study. 

A) The abstract is sufficiently developed, and it is useful to frame the purpose of the study, but a few concerns are present:

Comment 1: It would be advisable to make a good introduction to the manuscript before the aim of the study.

Comment 2: It would be appropriate to specify the characteristics of the study.

B) In the introduction, the characteristics of the advances in anterior cruciate ligament reconstruction have been sufficiently described. 

Comment 3For the sake of completeness, it would be advisable to add some information regarding the management of the advances in anterior cruciate ligament reconstruction in the young populationAdding some bibliographic references about it, for example: (Shumborski S, et al (2020). "Allograft Donor Characteristics Significantly Influence Graft Rupture After Anterior Cruciate Ligament Reconstruction in a Young Active Population")

Commnet 4: “However, to the best of our knowledge, there is a paucity in literature regarding the influence of interference-screw position on postoperative TE and postoperative clinical results. Considering the significant role of the ACL in the anterior and posterior stabilization of the knee, these two screw different positions might influence the  biomechanical relationship between graft and bone tunnel, leading to differences in postoperative TE and clinical outcomes.” Please adding some bibliographic references.

C) The materials and methods and results have been adequatelydeveloped.

Comment 5Why did you not consider in depth the return to sport in your results? it would have been interesting to analyze recovery times.

D) The discussion is sufficiently developed.

Comment 6You might add something to the conclusions about “Posterior screw position in the tibial tunnel maybe a better choice

Finally, English language editing is needed.

Nevertheless, some minor changes are needed to be considered suitable for publication.

Author Response

Thank you very much for your valuable comment. We appreciate the time and effort dedicated by you. The comments provided were valuable and helped us refine our paper. Attached Word are our point-by-point responses to your comments.

Reviewer 2 Report

Thank you very much for the opportunity to review interesting research. I have a few comments:

1. Who qualified the patients for the study and who divided them into two groups?

2. Are proprioception exercises included in the physiotherapy program? If so, what exercises, and if not, why?

3. Who performed the Lachman test and what test methodology was adopted?

4. Please include information about the improvement of functions in the conclusions - specifically what indicators have improved.

5. Please update the references to be no more than 10 years old

Author Response

Thank you very much for your valuable comment. We appreciate the time and effort dedicated by you. The comments provided were valuable and helped us refine our paper. Attached Word is our point-by-point responses to your comments.

Round 2

Reviewer 2 Report

The authors responded to all the comments in a sufficient way, which is why I agree with the publication of research.